# Concentric Circles: A New Ultrasonographic Sign for the Diagnosis of Normal Infantile Hip Development

**DOI:** 10.3390/children10010168

**Published:** 2023-01-15

**Authors:** Nikolaos Laliotis, Chrysanthos Chrysanthou, Panagiotis Konstandinidis

**Affiliations:** Orthopaedic Department, Inter Balkan Medical Center, Asklipiou 10 Pilea, 57001 Thessaloniki, Greece

**Keywords:** hip ultrasound, developmental dysplasia of the hip, concentric circle sign, hip dysplasia, diagnostic imaging

## Abstract

Ultrasound (US) of the infant hip is used to diagnose developmental dysplasia of the hip (DDH). We present a new sonographic sign that describes the periphery of the femoral head and the acetabulum as two concentric circles.During 2008–2019, 3650 infants were referred for diagnosis of DDH. All underwent a clinical and US examination. We recorded the femoral head as the inner circle, within a fixed external circle, which was identified as the acetabulum. We analysed the clinical signs and risk factors. The US sign of two concentric circles was normal in 3522 infants and was classified as normal hip development. The alpha angle was >60° in 3449 (95%) infants. For the remaining 73 (5%) infants, the alpha angle was 50–60° and underwent further follow-up examination until the alpha angle was normalised. In 128 babies (3.5%), we detected the disruption of the concentric circle sign; the femoral head was found outside the acetabulum, which appeared with an upward sloping roof and the alpha angle was <50°. These infants had DDH and received appropriate treatment. Infants with a concentric circle sign and normal alpha angle are normal, whereas those with a disrupted sign are considered as having DDH.

## 1. Introduction

Developmental dysplasia of the hip (DDH) is the most common disorder of the infant hip. A clinical examination is an essential part of diagnosis for infants with potential hip dysplasia. The main clinical features are a positive Ortolani sign, reduced hip abduction, and apparent leg length discrepancy. A variety of risk factors for hip dysplasia have been reported and the most common include breech presentation, positive family history, and multiple pregnancies. Congenital foot abnormalities, congenital muscular torticollis, prematurity, low birth weight, and oligohydramnios are among other risk factors [1,2,3]. A definite diagnosis for the normal or dysplastic hip is established with ultrasound examination (US) and, if required, is followed by an X-ray examination [4,5,6]. Hip joint sonography, as described by Graf, describes the hip anatomy with an area of immature hip development between that of the normal and dysplastic hip [7].Rosendahl et al. described hip stability during the US examination using the term “concentric” with the dynamic Barlow manoeuvreand measured the gap between the femoral head and the acetabulum. She proposed the division of the infant hip into four types as stable, minorinstability, major instability, and dislocated hips [8,9].

Our study presents a new sign during sonography of the infantile hip. We recorded in the standard coronal plane, the spherical periphery of the femoral head lying inside the concentric spherical acetabulum, which forms a double concentric circle figure. When hip development is abnormal, as in dysplasia or dislocation, this sign is not detected. The acetabulum appears as an elliptic surface, there is clear disruption of the concentric circle sign, and the head appears as a sphere in a sloping roof.

## 2. Patients and Methods

### 2.1. Study Group

A cohort of clinical and ultrasound (US) data were prospectively collected during the decade 2008–2019. In our paediatric orthopaedic department, 3650 neonates and infants were referred from their attending paediatricianfor a hip evaluation. The infants presented with suspicious clinical signs or risk factors for hip dysplasia. Their ages ranged from 10 days to 12 months. This selective cohort of patients was specifically referred from their community or hospital paediatricians and did not represent a screening cohort.

Initially, we formulated an evaluation data collection form to record the sex, name and date of birth, and aetiology of the referral. We recorded as risk factors the positive family history for hip dysplasia, breech presentation, twins, or multiple birth. We have not evaluated other risk factors such as prematurity or low birth weight.

Positive family history referred to babies with a first-degree relative that required an intervention, such as orthotic treatment or surgery, excluding a history of double nappies.

### 2.2. Clinical Examination

Clinical signs suspicious for DDH were reduced hip abduction (bilateral or unilateral), asymmetry of gluteal creases, leg length discrepancy, or pelvic tilt. The asymmetry of gluteal creases was also recorded in the prone position. Infants with foot disorders and torticollis were included in the group of children that were referred from the clinical examination.

Ortolani and Barlow tests were used to evaluate the stability of the infant’s hip. We recorded the test as negative or positive, indicating the possible presence of dysplasia. We were careful with the strength used for performing the test, especially in neonates during the first month of age. Infants referred for clicky hips were evaluated in the clinical examination and scored only as showing positive or negative Ortolani test findings.

### 2.3. Ultrasound Examination

US examination was performed during the same referral examination. We performed the scans according to Graf’s principles, trying to identify the lower limb, iliac bony rim, and prominent acetabular labrum. Examination was performed along the coronal plane. We measured the alpha angle. US was performed using a GE Logiq 100 system with a 7.5-MHz linear transducer (GE-Healthcare, Milwaukee, IL, USA).

We defined the periphery of the femoral head and the subchondral part of the acetabulum. The acetabulum was recorded from the labrum to the triradiate cartilage, at the boundary of the growth plate of the femoral head. Thesigns formed an image resembling double concentric circles, clearly incorporated one in the other. The observation was clear, and it was not necessary to draw lines. The presence of the ossification nucleus was recorded and compared to the size in each leg.

A dynamic US examination followed, which was performed by internal and external rotation of the limbto evaluate the movement of the inner circle, the femoral head, while the external circle, the acetabulum, remained rigid. The inner circle was rotated without losing contact with the acetabulum. In the presence of dysplasia or dislocation, the round femoral head was detected but the acetabulum appeared with an upward sloping roof and we could not define its spherical shape. The inner acetabulum presented an elliptic shape, and the round femoral head was at a distance from the triradiate cartilage. A spherical femoral head lying on an elliptically shaped acetabulum could be detected. For all unilateral cases, when the ossific nucleus appeared on the normal side, there was a distinct difference in the size of the affected nucleus in contrast to the normal nucleus. The average time required for the entire evaluationin calm babies was 10 min (Figure 1, Figure 2, Figure 3, Figure 4 and Figure 5).

## 3. Results

Of the 3650 babies included in our study, 2336 (64%) were female and 1314 (36%) were male. The age at their initial examination varied, with a higher prevalence at the ages of 3–5 months.

Of the 3650 infants, 2628 (72%) were referred because of suspicious clinical signs, while the remaining 1022 (28%) were referred due to the presence of risk factors. Of these, 475 babies (13%) had been referred for both clinical signs and risk factors.

Overall, 295 babies in the group with clinical signs also had risk factors for DDH and 180 babies in the group of infants with risk factors also had clinical findings suspicious for DDH.

Risk factors for DDH included babies born in breech presentation, the presence of positive family history, and multiple pregnancies (Table 1).

Infants referred after the clinical examination involved a total of 2808 (2628 suspected infants +180 infants with clinical signs), of these 1798 (64%) babies presented with reduced abduction of the hips affecting either both hips (85%) or a unilateral hip (15%). Leg length discrepancy and pelvic tilt were the referral aetiology in 56 babies (2%), and foot abnormalities and torticollis were those in 112 babies (4%). Asymmetry of the gluteal creases was observed in 168 babies (6%) (Table 2).

Of note, 674 infants were also referred to our clinic because of positive Ortolani and Barlow test results, including ‘clicky hips’. Of these, in clinical examination, 95 children were found to be positive for the manoeuvre.

Overall, 3650 babies underwent US examination.

Using the criteria of the two concentric circles, 3522 (96.5%) babies presented normal hips. Among them, 3449 (94.5%) had an alpha angle >60° and 73 (2.0%) had an alpha angle between 50° and 60° and were classified as having immature hip. Based on the concentric circle sign, we identified contact and smooth movement of the head in the acetabulum, but these children were assigned to further evaluation, until the appearance of the ossific nucleus and normalisation of the alpha angle. In 128 infants (3.5%), there was a clear disruption of the two concentric circles and the hip was classified as subluxated or dislocated. The Graf classification was types threeand four, with an alpha angle <50°. Of these, bilateral dysplasia was detected in 18 babies. The children were treated, either using a modified Pavlik or Tubingen brace when aged <3 monthsor with closed reduction under anaesthesia and arthrogram, if not easily reduced.

In the group of 128 children diagnosed with DDH, only eight were males. Among them, 124 children had a positive clinical sign of reduced abduction, positive Ortolani test, andapparentleg length discrepancy (LLD). The remaining four infants had a normal clinical evaluationbut had been referred for breech presentation with loose broad abduction of the hips.

## 4. Discussion

We describe a new sign on US examination of the infantile hip. Ultrasound is an essential approach to examine hips in the first year of birth. Graf provided a clear definition for accurate assessment of the hip, emphasising the exact plane of examination with the lower limb, iliac bone, and upper part of the labrum. Measurement of the alpha angle describes the osseous part of the acetabulum. Despite considerable variability, alpha angle measurements have been described to be associated with age, sex, and side [4,7,10]. We adopted the initially defined alpha values for normal as >60°. Nonetheless, measurements of the beta angle are not always accurate. Graf divided neonatal hips into four groups and further subdivided type II into subgroups a, b, and c in order to deal with the immature group of neonatal hips. The reliability of the Graf classification has been questioned several times, as it depends on various subjective parameters, including the experience of the examiner and orientation of the US transducer, but mainly because it requires an accurate plane of examination [11,12,13,14].

Harcke et al. utilised the dynamic measurements of the neonatal hip, while Terjersen and Morin measured the coverage of the femoral head, as a percentage of the height of the femoral head is covered by the osseous part of the acetabulum [15,16,17,18]. Hosny et al. proposed a new angle for measurement, which combined the alpha and beta angles [19]. All these measurements are well established in the medical literature.

Several authors have tested the reproducibility of US tests and defined the alpha angle as the most accurately reproducible measurement. These measurements were mainly tested in neonates, while in our cohort, mainly older infants were examined. As neonates grow, it becomes easier to differentiate the immature normal hip from a dysplastic hip. Peterlein et al. described the presence of an angled or round bony roof. In our study, this was considered a normal finding when the femoral head produced the concentric circle sign with the acetabulum and had a normal alpha angle [20,21].

Rosendahl et al. described the dynamic test by dividing neonates into four groups. The authors used the term “concentricity” and the main sign during the test was evaluation of the gap between the femoral head and the acetabulum, combined with possible displacement of the labrum. They described minor or major instability by measuring the gap in the acetabular depth [8,22].

During the first week of life, hip instability with a positive Ortolani sign is common and usually improves. The strength required to perform tests of stability may overestimate the normal laxity and possibly the diagnosis of neonatal hip dysplasia, leading in overtreatment of the normal neonatal hip.

The concentric double circle sign is easily identified either in static or dynamic US examination. The periphery of the femoral head rotates but remains in contact with the acetabulum in normal hips. We did not draw circles to figure these circles as we found it easy to estimate their concentricity. However, accurate digital measurements for the sphericity of the two circles can be made during US, similar to Mose measurements of the sphericity of the femoral head in Perthes’ disease.

All hips that were measured with an alpha angle >60° of Graf type I presented with a normal concentric circles sign. Hips with a normal concentric circle sign, but with an alpha angle of 50–60°, were classified as immature. They were regularly followed up to evaluate the normal development of the alpha angle in order to exclude hip dysplasia. An immature hip is expected to become normal by the second month of age, but a grey zone always exists. Hips characterized as immature may resolve spontaneously without any treatment [20,23,24].

All hipswith an alpha angle <50° indicated a clear disruption of the two concentric circles sign. In hip dislocation, the femoral head was found on the outer part of the ilium, while in subluxation, it was found on the edge of the sloping roof of the acetabulum. All these cases were further treated with orthotics using a modified Pavlik or Tubingen brace or by closed reduction under anaesthesia.

Clinical and US examination was performed simultaneously by the same paediatric orthopaedic surgeon. Usually, referrals to orthopaedic surgeons follow previously performed US examinations from trained neonatologists, radiologists, or sonographers. Ultrasonography is used to establish the diagnosis of hip dysplasia. The incidence of DDH is increased when the diagnosis is based purely on US examination [23,24].

All children with a positive Ortolani sign, when accurately performed and not mixed with a normal click, were classified as dysplastic and showed a disruption of the concentric double circle sign [25,26].

The disparity between clinical and US diagnosis in neonatal screening is well presented by Kuyng et al. [27]. They described clinical instability under the Barlow and Ortolani tests, referring to five types, including noisy hips and reported that 92% of subluxable hips were Graf I or IIa subtypes, which is a common finding in neonates with hip instability. They also reported that 73% of the dislocating or dislocated hips were Graf I or IIa subtypes on static examination. This differs from our result that all babies with a positive Ortolani finding presented with a disrupted concentric circle sign. The age of our patients and performance of the test by an experienced surgeon are possible explanations for this discrepancy.

Infants referred for minor limitation of hip abduction are generally found to be normal on US examination. Similar findings were observed in our patients. However, in infants with bilaterally dislocated hips, limited abduction may be the only important clinical sign, with the absence of instability. In these patients, US examination confirmed the clear disruption of the concentric circle sign, with the femoral head lying outside the sloping acetabulum.

The increased incidence of DDH in our patients is justified as the selected babies were referred due to positive clinical findings and risk factors but were not cases derived from a general screening test.

## 5. Conclusions

We present a new ultrasonographic sign that presents the femoral head and the acetabulum as two concentric circles. The sign can be reproduced both in the static and dynamic examination of the hip. It can be used in combination with alpha angle measurement to diagnose the normal development of the infant’s hip. The sign is disrupted in infants with DDH. Infants defined as having immature hips according to the Graf classification should be followed up until confirmation of normal hip development is established.

## Figures and Tables

**Figure 1 children-10-00168-f001:**
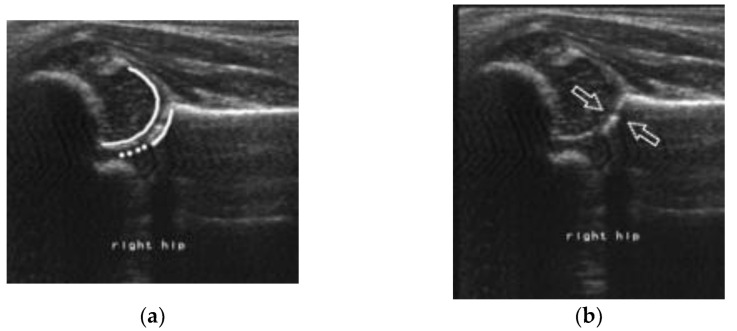
Representative image of a concentric double circle. (**a**): The inner circle is the femoral head and the outer circle is the acetabulum. (**b**): The arrows point to the boundaries of the head and the acetabulum.

**Figure 2 children-10-00168-f002:**
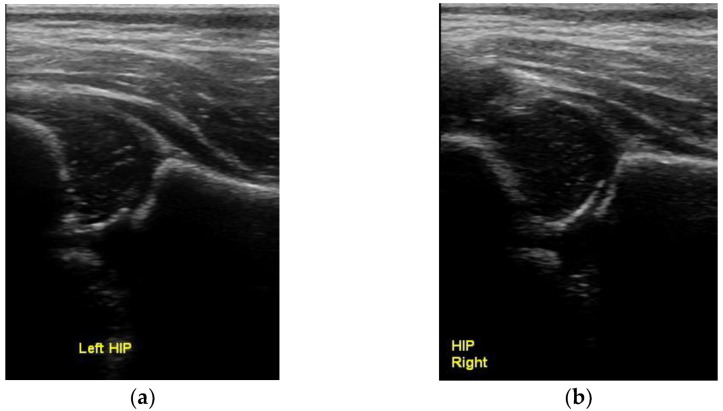
Normal concentric circles with the spherical femoral head lying inside the spherical subchondral acetabulum. (**a**) left and (**b**) right hip.

**Figure 3 children-10-00168-f003:**
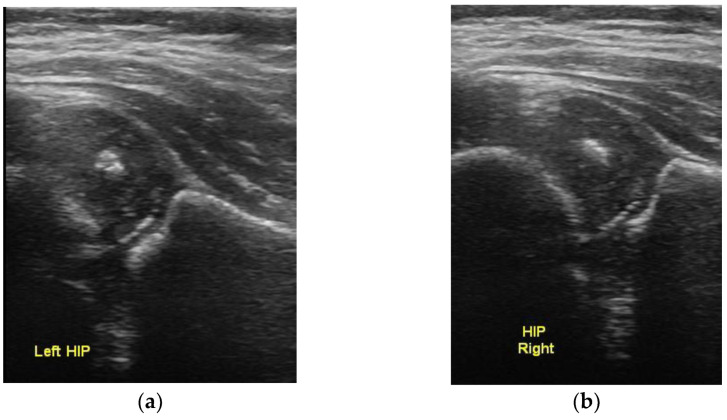
Bilateral normal ultrasound with the presence of ossific nuclei of the femoral head, showing the normal concentric circles sign. (**a**) left and (**b**) right hip.

**Figure 4 children-10-00168-f004:**
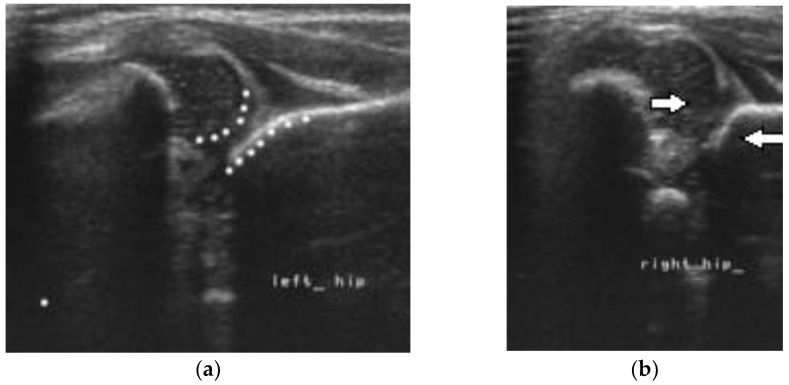
Bilateral DDH with disruption of the double concentric sign on the left and right side. (**a**) The acetabulum is elliptic and the head has a spherical shape (**b**) The arrows point to the femoral head and the sloping acetabulum. There is increased distance between the head and the depth of the acetabulum.

**Figure 5 children-10-00168-f005:**
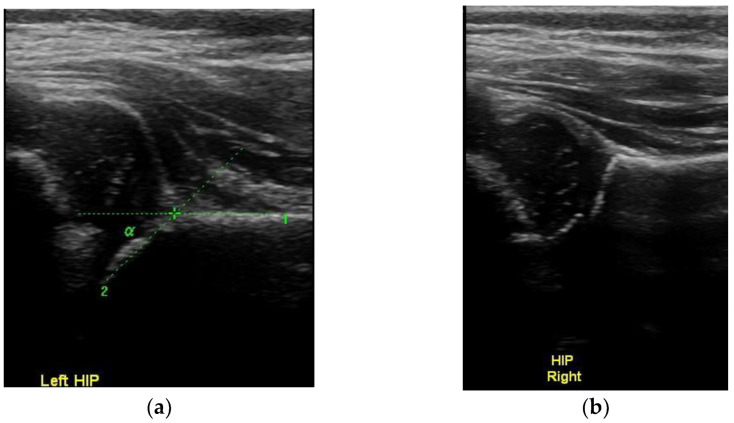
Image showing DDH (**a**) on the left hip, with the spherical head lying on the upper part of the elliptic shape acetabulum and (**b**) the normal right hip.

**Table 1 children-10-00168-t001:** Reasons for referral to the orthopaedic unit.

	Numberof Infants	Percentage %
Clinical Examination	2628	72%
2. Risk Factors a. Positive family history b. Breech Presentation c. Multiple babies	1022 a. 396 b. 816 c. 105	28% a. 30% b. 62% c. 8%
3. Combined etiology	475	13%

**Table 2 children-10-00168-t002:** Results of clinical examination of referred infants for DDH (*n* = 2808).

	Number of Infants	Percentage (%)
Reduced Abduction	1798	64%
LLD and Pelvic Tilt	56	2%
Asymmetry of Creases	168	6%
Ortolani Positive	674	24%
Foot Abnormalities	112	4%

## Data Availability

Data are available upon request from the corresponding author.

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
