# Peer review of "Concentric Circles: A New Ultrasonographic Sign for the Diagnosis of Normal Infantile Hip Development"

_children, 2023, doi:10.3390/children10010168_

Round 1

Reviewer 1 Report

First I would like to congratulate the authors on an interesting research article.

The formal outline of the article is out of the template and the text doesn’t continue in order, but these are only cosmetic issues.

Introduction

-          Abstract and introduction are separate parts of the article, so abbreviations have to be explained in both when they first appear –> US in the introduction and not in patients and methods

-          Citing style in the text – refer to citations, as they are mentioned in the text after individual sentences. Ending with (1-7) references is not appropriate, please revise the whole article.

-          The introduction needs to be larger, and include information about etiology, prevalence, genetics, and risk factors – for example, find and use/cite articles available within the publishers' repertoire.

-          Line 63 “we recorded” – also later are similar instances present = whole text needs to be revised for spelling and clarity

Patients and method

-          Try to divide the text into some concise parts/sections e.g. – study group (with inclusion criteria, …), Clinical examination, optional US examination, statistical analysis

Results

-          Add percentual values to patient counts.

-          Fuse table 1 and table 2 as they are virtually obsolete separately

-          Revise and reorder the whole results section as now it is really distracting and difficult to read

Discussion

-          Same citation issue as the introduction. Indeed multiple citations are suitable where they should be, please revise.

-          Include comments on risk factors present in the study in the discussion section

References

-          Many references are very old, update these if they aren’t vital to your work

Informed consent issue – "parents were informed" It doesn’t mean parents agree, it would be beneficial to address this issue if the Review Board had any comments, etc.

Author Response

Dear reviewer no 1

 We would like to express our sincere gratitude for your effort and time to evaluate our manuscript.

We really thank you for your comments

We have corrected our manuscript according to your suggestions

 Introduction: abbreviations and citations were corrected, risk factors were added.

The language format was corrected from the beginning from Editage (editing services). We used the word record in present tense but we agree that past tense in most appropriate

Patients and method were divided in groups.

Results were corrected and tables were moved accordingly.

Citations in discussion were changed.

 Regarding your recommendation for comments on risk factors in DDH, we would like to emphasize that this manuscript is not focused on DDH in general terms, but it describes the new US sign of concentric circles, that is simultaneously performed from the Pediatric Orthopaedic Surgeon, in combination with the clinical examination.  We are completing our study on the relation of foot disorders and DDH in infants that is soon going to be submitted for evaluation.

References are related with US examination, up to 2020. We have used references that are important for the evaluation of ultrasonography in infants.

Yours sincerely

Nikolaos Laliotis

Pediatric Orthopaedic Surgeon

M.Ch.Orth

Former Assistant Professor in Pediatric Orthopaedics

Reviewer 2 Report

Very interesting paper with a new ultrasonographic sign that presents the femoral head as two concentric circles. I think it would be useful for radiologist and pediatrician and pediatric orthopedic surgeon

Author Response

Dear reviewer no 2

 We would like to express our sincere gratitude for your effort and time to evaluate our manuscript.

We are DELIGHTED from your comments on our work, we really thank you.

Yours sincerely

Nikolaos Laliotis

Pediatric Orthopaedic Surgeon

M.Ch.Orth

Former Assistant Professor in Pediatric Orthopaedics

Inter Balkan Medical Center Thessaloniki Greece

Reviewer 3 Report

Dear Author,

Thank you for the opportunity to review this article.

It gathers an impressive patient pool in 10 years of medical activity, with thorough results.

Statistical analysis could be done a bit better in order to correlate your novel sign with alpha angle intervals, not only with the threshold of 50 degrees which may better correlate with the prognosis rather than the known classification system (Graf IIC is also centered). Were all the children treated accordingly with complete restitution?

There are more risk factors of DDH known in the literature and more associations and you should publish research work about finding additional correlations. Here is an example of an effort to find new correlations to already known risk factors in newborn orthopedic pathology: Obstetric fractures in cesarean delivery and risk factors as evaluated by pediatric surgeons, published in International Orthopaedics, DOI 10.1007/s00264-022-05547-2.

This article is prone to publication after a minor revision.

Author Response

Dear reviewer no 3

 We would like to express our sincere gratitude for your effort and time to evaluate our manuscript. We are DELIGHTED from your comments on our work, we really thank you.

According to your suggestions, we have corrected the part of risk factors in our paper. We would like to emphasize that this manuscript is not focused on DDH in general terms, but it describes the new US sign of concentric circles, that is simultaneously performed from the Pediatric Orthopaedic Surgeon, in combination with the clinical examination.  We are completing our study on the relation of foot disorders and DDH in infants that is soon going to be submitted for evaluation.

 Regarding the patients that are in the grey zone of Graf II B and C, our sign is used in combination with a angle and not as a single sign. We made clear in the results that we follow up these children until the normalization of a angle and the appearance of the ossific nucleus.

 We also report that children with disrupted concentric circles and a angle< 50d, were appropriately treated either with a frame or closed reduction under anesthesia and arthrogram.

Yours sincerely

Nikolaos Laliotis

Pediatric Orthopaedic Surgeon

M.Ch.Orth

Former Assistant Professor in Pediatric Orthopaedics

Inter Balkan Medical Center Thessaloniki Greece

Round 2

Reviewer 1 Report

The article was improved in language and content-wise. I also get the points about the conclusion and risk factors. However, in the Introduction, no citations on risk factors were included as I suggested in my comments which I find essential to be corrected. So, as I suggested revise the introduction accordingly, for example using 2-3 MDPI sources to do so. Also, tables 1 and 2 were not merged which I also think is necessary. Insert table 2 into table 1 as sub-columns under Risk factors, this will enhance the clarity of the message.

Author Response

Dear reviewer 1, once again we thank you for your suggestions, your time and effort to evaluate and further improve the manuscript.

1 We have included 3 references in the introduction, relevant to risk factors and we have corrected the numerical appearance of all references

Introduction

Developmental dysplasia of the hip (DDH) is the most common disorder of the infant hip. A clinical examination is an essential part of diagnosis for infants with potential hip dysplasia. The main clinical features are a positive Ortolani sign, reduced hip abduction, andapparent leg length discrepancy. A variety of risk factors for hip dysplasia has been reported and most common include breech presentation, positive family history, and multiple pregnancies. Congenital foot abnormalities, congenital muscular torticollis, prematurity, low birth weight, oligohydramnios are among other risk factors. [1-3]  A definite diagnosis for the normal or dysplastic hip is established with ultrasound examination (US) and, if required, is followed by an X-ray examination.[4-6] Hip joint sonography, as described by Graf, describes the hip anatomy with an area of immature hip development between that of the normal and dysplastic hip.[7]  Rosendahl et al. described hip stability during the US examinationusing the term “concentric” with the dynamic Barlow manoeuvreand measured the gap between the femoral head and the acetabulum. She proposed the division of the infant hip into four types as stable, minorinstability, major instability, and dislocated hips [8].

References

[1] Roposch A, Protopapa E, Malaga-Shaw O, Gelfer Y, Humphries P, Ridout D, Wedge JH. Predicting developmental dysplasia of the hip in at-risk newborns. BMC Musculoskelet Disord. 2020 Jul 7;21(1):442. doi: 10.1186/s12891-020-03454-4. PMID: 32635922; PMCID: PMC7341560

[2] Oh EJ, Min JJ, Kwon SS, Kim SB, Choi CW, Jung YH, Oh KJ, Park JY, Park MS. Breech Presentation in Twins as a Risk Factor for Developmental Dysplasia of the Hip. J Pediatr Orthop. 2022 Jan 1;42(1):e55-e58. doi: 10.1097/BPO.0000000000001982. PMID: 34619721; PMCID: PMC8663528.

[3] Harsanyi S, Zamborsky R, Krajciova L, Kokavec M, Danisovic L. Developmental Dysplasia of the Hip: A Review of Etiopathogenesis, Risk Factors, and Genetic Aspects. Medicina (Kaunas). 2020 Mar 31;56(4):153. doi: 10.3390/medicina56040153. PMID: 32244273; PMCID: PMC7230892.

2  we have merged tables 1 and 2 according to your suggestion

Reasons for referral to the orthopaedic unit.

Number of infants

Percentage %

1.      Clinical Examination

2628

72%

2.      Risk Factors

a.      Positive family history

b.      Breech Presentation

c.      Multiple babies

1022

a.      396

b.      816

c.      105

28%

a.      30%

b.      62%

c.      8%

3.      Combined etiology

475

13%

3 Once again, we thank you for your nice comments and your effort

Yours sincerely

Nikolaos Laliotis

Pediatric Orthopaedic Surgeon 

M.Ch.Orth

f Assistant Professor in Pediatric Orthopaedics

Inter Balkan Medical Center Thessaloniki Greece
